# Blocking Periostin Prevented Development of Inflammation in Rhabdomyolysis-Induced Acute Kidney Injury Mice Model

**DOI:** 10.3390/cells11213388

**Published:** 2022-10-27

**Authors:** Jun Muratsu, Fumihiro Sanada, Nobutaka Koibuchi, Kana Shibata, Naruto Katsuragi, Shoji Ikebe, Yasuo Tsunetoshi, Hiromi Rakugi, Ryuichi Morishita, Yoshiaki Taniyama

**Affiliations:** 1Department of Clinical Gene Therapy, Osaka University Graduate School of Medicine, Suita 565-0871, Japan; 2Department of General and Geriatric Medicine, Osaka University Graduate School of Medicine, Suita 565-0871, Japan; 3Department of Advanced Molecular Therapy, Osaka University Graduate School of Medicine, Suita 565-0871, Japan; 4Second Department of Oral and Maxillofacial Surgery, Osaka Dental University, Osaka 540-0008, Japan

**Keywords:** periostin, acute kidney injury, rhabdomyolysis, inflammation

## Abstract

Background: Rhabdomyolysis is the collapse of damaged skeletal muscle and the leakage of muscle-cell contents, such as electrolytes, myoglobin, and other sarcoplasmic proteins, into the circulation. The glomeruli filtered these products, leading to acute kidney injury (AKI) through several mechanisms, such as intratubular obstruction secondary to protein precipitation. The prognosis is highly mutable and depends on the underlying complications and etiologies. New therapeutic plans to reduce AKI are now needed. Up to now, several cellular pathways, with the nuclear factor kappa beta (NF-kB), as well as the proinflammatory effects on epithelial and tubular epithelial cells, have been recognized as the major pathway for the initiation of the matrix-producing cells in AKI. Recently, it has been mentioned that periostin (POSTN), an extracellular matrix protein, is involved in the development of inflammation through the modulation of the NF-kB pathway. However, how POSTN develops the inflammation protection in AKI by rhabdomyolysis is uncertain. This study aimed to investigate the role of POSTN in a rhabdomyolysis mice model of AKI induced by an intramuscular injection of 50% glycerol. Methods: In vivo, we performed an intramuscular injection of 50% glycerol (5 mg/kg body weight) to make rhabdomyolysis-induced AKI. We examined the expression level of POSTN through the progression of AKI after glycerol intramuscular injection for C57BL/6J wildtype (WT) mice. We sacrificed mice at 72 h after glycerol injection. We made periostin-null mice to examine the role of POSTN in acute renal failure. The role of periostin was further examined through in vitro methods. The development of renal inflammation is linked with the NF-kB pathway. To examine the POSTN function, we administrated hemin (100 μM) on NIH-3T3 fibroblast cells, and the following signaling pathways were examined. Results: The expression of periostin was highly increased, peaking at about 72 h after glycerol injection. The expression of inflammation-associated mRNAs such as monocyte chemotactic protein-1 (MCP-1), tumor necrosis factor-alpha (TNF-a) and IL-6, and tubular injury score in H-E staining were more reduced in POSTN-null mice than WT mice at 72 h after glycerol injection. Conclusion: POSTN was highly expressed in the kidney through rhabdomyolysis and was a positive regulator of AKI. Targeting POSTN might propose a new therapeutic strategy against the development of acute renal failure.

## 1. Introduction

Chronic kidney disease (CKD) patients hold a high risk of end-stage kidney disease (ESKD) [1], cardiovascular disease (CVD) [2], and mortality [3]. CKD is caused by various factors such as age [4,5], lifestyle [6,7,8], obesity [9,10], type 2 diabetes [11], metabolic syndrome [12], hypertension [13], and acute kidney injury. Acute kidney injury as a complication of rhabdomyolysis is quite common, representing about 7 to 10% of all causes of acute kidney injury in the USA [14,15]. Rhabdomyolysis is a condition induced by skeletal muscle injury that usually promotes acute kidney injury (AKI). Rhabdomyolysis has been linked to different conditions, including severe trauma and intense physical exercise. Statins, a group of drugs used for the treatment of hypercholesterolemia, have adverse effects on skeletal muscle and induced rhabdomyolysis [16]. Myoglobin-induced renal toxicity increases inflammation, oxidative stress, vasoconstriction, endothelial dysfunction, and apoptosis in rhabdomyolysis-associated kidney damage [17]. New drugs to prevent rhabdomyolysis-induced AKI are now needed.

Periostin is highly expressed in chronic inflammatory diseases, including asthma [18], allergic conjunctivitis, eosinophilic chronic sinusitis/chronic rhinosinusitis with nasal polyp, and atopic dermatitis [19], and periostin plays important roles in the pathogenesis of these diseases. Some reports revealed the association between periostin and acute kidney disease in the renal ischemia-reperfusion injury model [20]. However, many points remains unknown about the role of periostin in the acute kidney disease. The epithelial/mesenchymal interaction via periostin is important for the onset of allergic inflammation, in which periostin derived from fibroblasts acts on epithelial cells or fibroblasts, activating their NF-κB [21]. Now, the significance of periostin has been expanded into other inflammatory or fibrotic diseases such as scleroderma and pulmonary fibrosis.

Recently, new therapeutic agents for CKD such as dapagliflozin [22], finerenone [23] have been developed. However, few new treatments for AKI have been developed. In the current study, we sought to define the impact of periostin for rhabdomyolysis-induced AKI.

## 2. Methods

### 2.1. Ethical Statement

All the experimental methos were accepted by the Institutional Animal Committee at the Department of Veterinary Science of Osaka University School of Medicine and followed the recommendations of the guidelines for animal experimentation at research institutes (Ministry of Education, Culture, Sports, Science and Technology, Osaka, Japan), guidelines for animal experimentation at institutes (Ministry of Health, Labor and Welfare, Japan), and guidelines for proper conduct for animal experimentation (Science Council of Japan). Male C57BL/6J mice, approximately 8 weeks old, were purchased from Oriental Bio. Service, Japan. Periostin-null mice were previously established at RIKEN, Japan, and male mice aged approximately 8 weeks were used for the experiments.

We modeled rhabdomyolysis with a well-characterized model, glycerol intramuscular injection of 50% glycerol in normal saline, preceded by a period of water deprivation [24,25]. And we observed the serum creatinine kinase at 6hours later from glycerol injection, and the severe tubular damage in Hematoxylin-Eosin (H-E) staining at 72 hours later from glycerol injection.

We administered an intramuscular glycerol (5 mL/kg body weight) injection into mice to induce rhabdomyolysis-induced AKI. The day before injection, the mice abstained from drinking. On the day of injection, the mice were anesthetized by inhalation or intravenous injection, 50% glycerol (diluted with normal saline) was injected into the thigh muscle, and water deprivation was cancelled. The mice were sacrificed 72 h after glycerol injection (Figure 1). After sufficient anesthesia, the blood was obtained from the tail vein, and the mice were sacrificed.

### 2.2. Cell Cultures

In AKI induced by rhabdomyolysis, heme derived by myoglobin is instantly changed to hemin in the blood [26]. Hemin induces inflammation [27]. We performed an in vitro study using NIH3T3 fibroblast cells. The NIH3T3 fibroblast cells were cultured in DMEM + 10% FBS plus 1% penicillin/streptomycin. Removed the periostin gene from the NIH3T3 cells using genome editing technology. Fibroblasts were grown at 37 °C in a humidified 5% CO_2_ incubator and were serum starved for 24 h before stimulation with Hemin (100 μM) for 6 hrs (Figure 1). Using NIH3T3, periostin gene was removed by genome editing technology. We call them NIH3T3 PnKO.

### 2.3. Evaluation of Renal Histology

Formalin-fixed 4 μm paraffin-embedded sections were used for Hematoxylin-Eosin (H-E) staining. The measurement of tubular damage was analyzed by a BZ-II Analyzer (Keyence, Osaka, Japan). Tubular damage was defined by tubular epithelial necrosis, intratubular debris, and loss of brush border. The tubular damage score was 0: normal, 1: 1–25% damaged, 2: 26–50% damaged, 3: 51–75% damaged, and 4: 76–100% damaged. We examined 40 fields at random. Injury score (%) = (numbers of injured tubules/number of whole tubules) × 100.

### 2.4. Isolation of Total RNA and RT-PCR

Total RNA from mouse tissue was isolated using the RNeasy Mini Kit (QIAGEN, Hilden, Germany) according to the manufacturer’s instructions. DNase-treated total RNA was reverse-transcribed using the High-Capacity cDNA Reverse Transcriptase Kit (Applied Biosystems, Foster City, CA, USA) to produce complementary DNA. The cDNA encoding the target genes was amplified and quantified using a ViiA-7™ real-time PCR system (Applied Biosystems, Foster City, CA, USA) with the primer sets shown in Appendix A.

### 2.5. Statistical Analysis

All the statistical analyses were performed using the Stata, version 14.2 (Stata Corp., http://www.stata.com (accessed on 1 September 2022). We conducted statistical analyses between 17 January 2017 and 10 October 2022. The final analysis was performed on 10 October 2022), software package. Values are expressed as the means ± SEs. ANOVA and *t*-tests followed by the Tukey–Kramer adjustment for multiple comparisons were used to evaluate differences among more than two groups. In addition, linear regression analysis for the % tubular damage score (H-E staining) in the kidney was performed to clarify the association between % tubular damage and periostin expression in the kidney.

## 3. Results

### 3.1. Expression Profile of Periostin in Kidneys after 50% Glycerol Injection

First, the expression of periostin was measured by real-time PCR with samples obtained at day 0, 3, 21, and 42 after 50% glycerol injection in the rhabdomyolysis-induced acute kidney injury model. In this model, periostin expression was significantly elevated on day 21 compared with day 0. However, the upward tendency was observed from day 3. The mean expression of periostin in the kidney was peaking at approximately day 3 after 50% glycerol injection (Figure 2). 

Next, to elucidate the functional role of periostin in the rhabdomyolysis-induced acute kidney injury, we employed the rhabdomyolysis-induced acute kidney injury model in Pn-null mice. In the rhabdomyolysis, creatinine kinase was elevated from 4–6 h later from glycerol injection [28]. Serum creatinine kinase before injection and 6 h later from 50% glycerol injection, and creatinine at day 0 and day 3 after 50% glycerol injection were measured in wildtype and Pn-null mice. The levels of serum creatinine kinase was elevated in both WT and Pn-null rhabdomyolysis mice (Figure 3A). As shown in Figure 3B, in the wildtype mice, creatinine were elevated on day 3. The elevation of serum creatinine in Pn-null mice at day 3 was declined compared with WT at day 3. 

### 3.2. Genetic Deletion of Periostin Prevents the Development of Inflammation

Rhabdomyolysis-induced acute kidney injury is developed via several mechanisms, such as inflammation and intratubular obstruction secondary to protein precipitation. Hematoxylin-Eosin (H-E) staining in the wildtype mice rhabdomyolysis-induced acute kidney injury model showed tubular damage such as tubular epithelial necrosis, intratubular debris, and loss of brush border (Figure 4A). The tubular damage score examined using 40 fields taken randomly in the wildtype mice was higher than in the Pn-null mice (Figure 4B).

The expression of periostin and inflammation markers in the kidney are shown in Figure 5. All markers of inflammation were elevated in the wildtype mice on day 3 after 50% glycerol injection. On the other hand, all markers of inflammation declined significantly in the Pn-null mice on day 3 after 50% glycerol injection compared with WT. In addition, the expression of periostin in the kidney were significantly associated with % Tubular damage score (H-E staining) in the univariate linear regression analysis model (Appendix A). 

### 3.3. Expression of Inflammation Markers in NIH3T3 Fibroblasts after Hemin Stimulation

We demonstrated that periostin total knockdown similarly inhibited the downstream target transcripts of NF-kB in vitro. All the markers of inflammation declined in the periostin-knockdown NIH3T3 cells with hemin compared with the wildtype NIH3T3 cells with hemin (Figure 6).

## 4. Discussion

Previous reports suggested that periostin performs a crucial role in fibrosis, and acute kidney injury resulting in a high risk of progression to chronic kidney disease. Jung Nam An et al. reported periostin promoted kidney fibrosis through the p38 MAPK pathway following acute kidney injury triggered by a hypoxic or ischemic insult in a unilateral ischemia–reperfusion injury (UIRI) model [29]. However, the effect of periostin in a rhabdomyolysis-induced acute kidney injury model was unknown. The present study identified the impact of periostin for acute kidney injury induced by rhabdomyolysis.

In general, muscle cells damaged by rhabdomyolysis release immunostimulatory molecules, such as DNA, microRNAs, uric acid, and ligand high mobility group box proteins-1 (HMGB1). Activation of these inflammatory cells promotes the production of proinflammatory cytokines, such as tumor necrosis factor alpha (TNF-α) and transforming growth factor beta (TGF-β), thus resulting in the maintenance of a proinflammatory status [30,31].

Our study revealed that the expression of inflammation markers was significantly reduced in the Pn-null mice compared with the wildtype mice in the rhabdomyolysis-induced acute kidney injury model. Although the CK values were not significantly different between the WT and PN, the creatinine and tubular damage were significantly lower in the Pn-null than in the wildtype mice. This indicates that WT and Pn-null had sufficient muscle damage by glycerol injection. Furthermore, it shows that WT and Pn-null did not differ in muscle damage, but in periostin effects on the kidneys.

Rhabdomyolysis is an important clinical cause of AKI, due to the release of nephrotoxins (e.g., heme) from disrupted muscles [32]. A recent study showed that heme (hemin)-activated induced the macrophage extracellular traps (METs) and exacerbated rhabdomyolysis-induced acute kidney injury [33]. In rhabdomyolysis-induced AKI, myoglobin-derived heme is immediately converted to hemin in the blood [26]. Hemin induces multiple pathologic responses, such as cellular damage, inflammation, and endothelial activation, due to the ability to make reactive oxidative species [27].

In our study, after hemin stimulation, the expression of inflammation markers (i-NOS, MCP-1, TNF alpha, IL-6, and Interleukin-13 (IL-13)) was significantly reduced in the Pn-null compared with the wildtype mice. We demonstrated that periostin enhances iNOS expression and, subsequently, NF-κB signaling pathways. In previous reports, Periostin expression induced by oxidative stress contributes to myocardial fibrosis in high salt-induced hypertension model. They indicated that angiotensin II upregulated periostin and α-SMA expression compared with the control, whereas, pretreatment with N-acetyl-L-cysteine inhibited oxidative stress, periostin and α-SMA expression in fibroblasts [34]. Another reports revealed that the increase in periostin expression was correlated with the decrease in renal function, advanced stage renal damage and fibrosis, and NF-κB activation. Subsequently, Lilia Abbad et al. identified the novel pathways and genes regulated by the NF-κB-periostin interaction which are involved in the mechanisms of progression of Diabetic nephropathy [35]. The association between oxidative stress and AKI had been reported with various pathway [36,37]. Further investigation was expected.

There were several study limitations. We only studied the intramuscular injection of glycerol model, which lacked the functional readouts. The model also had no cardiovascular system risk factors, such as hypertension and diabetes. Other AKI models must be investigated to elucidate the clinical relevance of periostin in the future. In addition, we performed this study in only male mice. In the previous reports, one of the independent predictors for the rhabdomyolysis were female [38]. Therefore, after this study, the same study is required to be performed with female mice.

In conclusion, these findings indicate that periostin suppression might be a new therapeutic agent for acute renal injury caused by rhabdomyolysis.

## Figures and Tables

**Figure 1 cells-11-03388-f001:**
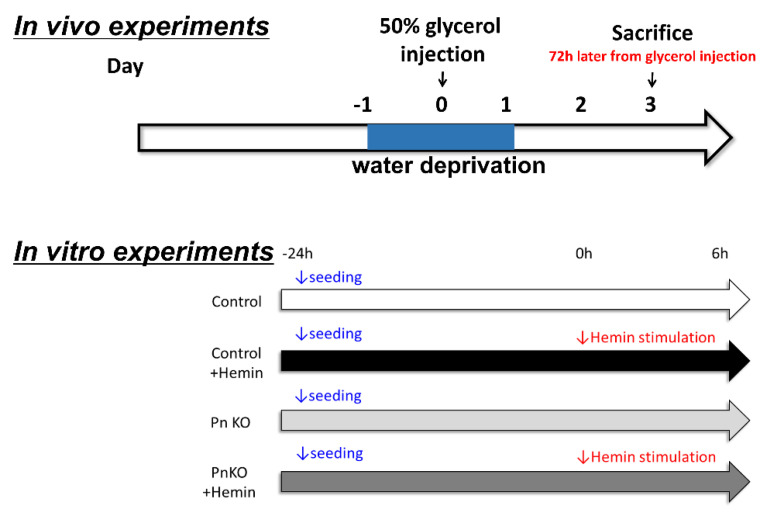
(In vivo experiments) The methods of the in vivo experiment schedule. 1. The day before the injection, the mice were abstained from drinking. 2. On the day of the injection, the mice were anesthetized by inhalation or intravenous injection, and 50% glycerol (diluted with normal saline) was injected into the thigh muscle. 3. The water deprivation was cancelled. 4. Acute renal failure due to rhabdomyolysis developed on day 3. (In vitro experiments) The methods of the in vitro experiment schedule. Cell: NIH3T3 fibroblast, medium: DMEM + 10% FBS, stimulus: Hemin (100 μM), stimulation time: 6 h, evaluation item: inflammation markers.

**Figure 2 cells-11-03388-f002:**
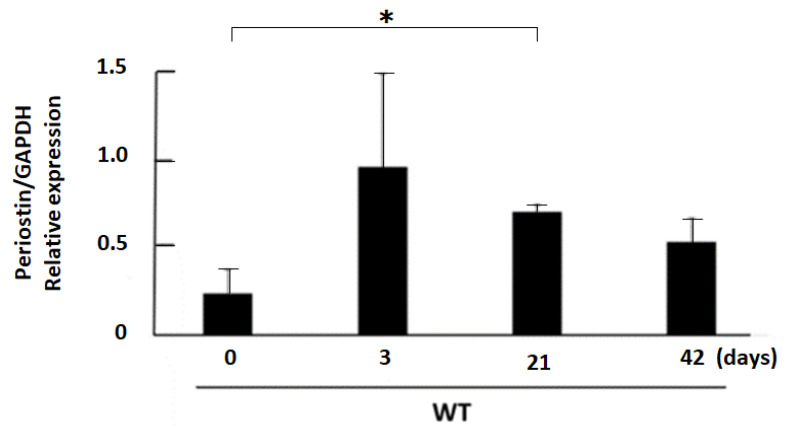
Periostin expression in the kidney of the rhabdomyolysis model. The expression of periostin in the kidney was highly increased, peaking at approximately day 3 after 50% glycerol injection. The relative gene expression was quantified at day 0–42 after operation by a real-time polymerase chain reaction using specific primers for periostin. * *p* < 0.05 vs. day 0. *n* = 3–4.

**Figure 3 cells-11-03388-f003:**
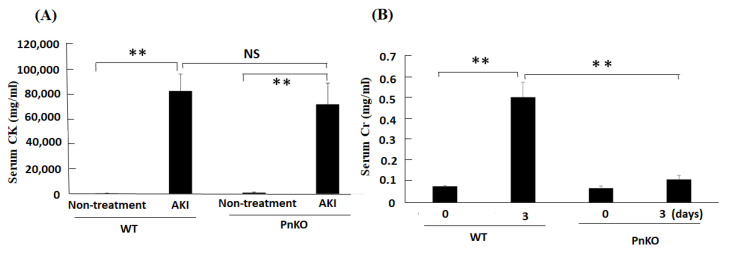
The levels of serum creatinine kinase (**A**) and creatinine (**B**) in wildtype and Pn-null rhabdomyolysis mouse model. (**A**) Serum creatinine kinase was quantified at 6 h later after glycerol injection. ** *p* < 0.05 vs. Non-treatment. *N* = 3–5. The levels of serum creatinine (**B**) was quantified at day 0 and 3 after operation. ** *p* < 0.05 vs. day 0, and day 3 (PnKO). *N* = 3–5.

**Figure 4 cells-11-03388-f004:**
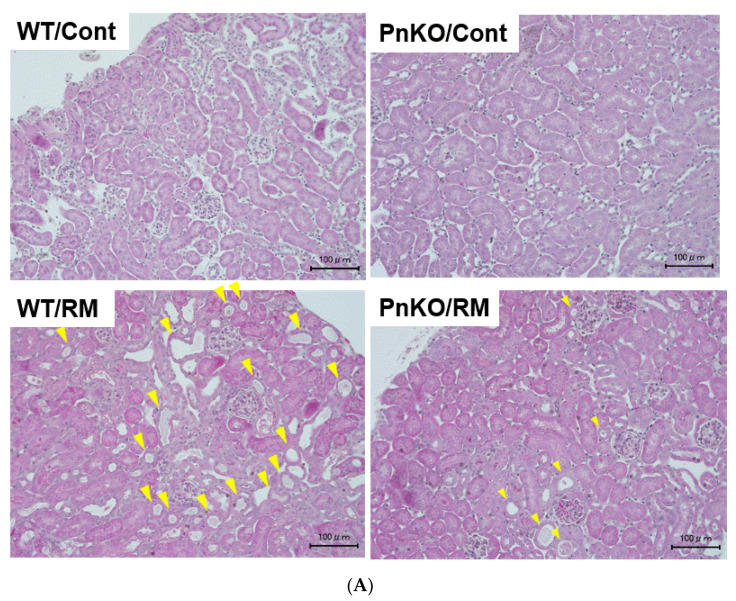
(**A**) The tubular damage in Hematoxylin-Eosin (H-E) staining (×100). Tubular damage in wildtype and Pn-null rhabdomyolysis mouse model kidney. Periostin-null mice exhibited reduced tubular damage in the kidney. (**B**) The evaluation of the degree of tubular damage in wildtype and Pn-null rhabdomyolysis mouse model kidney. The tubular damage was defined by tubular epithelial necrosis, intratubular debris, and loss of brush border. The tubular damage score was 0: normal, 1: 1–25% damaged, 2: 26–50% damaged, 3: 51–75% damaged, and 4: 76–100% damaged. We examined 40 fields at random. Injury score (%) = (numbers of injured tubules/number of whole tubules) × 100. ** *p* < 0.01 vs. day 0 (WT) and day 3 (PnKO) *n* = 6.

**Figure 5 cells-11-03388-f005:**
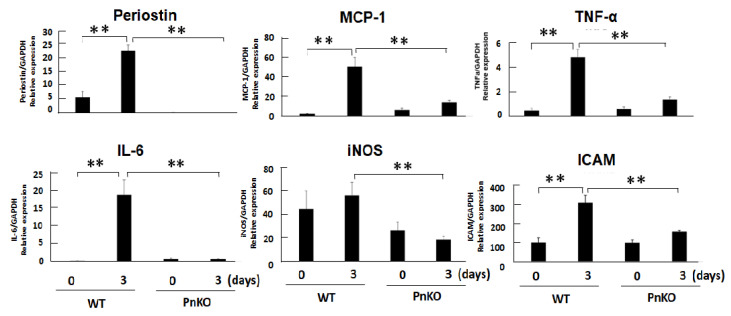
The expression of periostin and inflammation markers in the kidney of the rhabdomyolysis model. All markers of inflammation declined in the Pn-null rhabdomyolysis mouse model compared with the wildtype rhabdomyolysis mouse model. ** *p* < 0.01 vs. day 0 (WT) and day 3 (PnKO). *n* = 6.

**Figure 6 cells-11-03388-f006:**
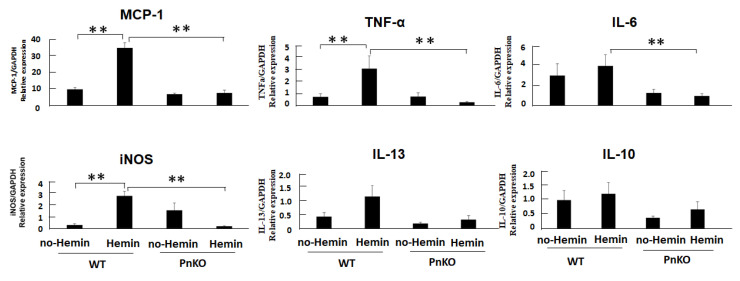
The expression of periostin and inflammation markers in the NIH3T3 fibroblasts after Hemin stimulation. All markers of inflammation declined in the periostin-knockdown NIH3T3 cells with hemin compared with the wildtype NIH3T3 fibroblasts with hemin. ** *p* < 0.05 vs. day 0 (WT) and day 3 (PnKO). *n* = 4.

## Data Availability

The data that support the findings of this study are available from the corresponding author, Y. Taniyama, upon reasonable request.

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
