# Peer review of "Blocking Periostin Prevented Development of Inflammation in Rhabdomyolysis-Induced Acute Kidney Injury Mice Model"

_cells, 2022, doi:10.3390/cells11213388_

Round 1

Reviewer 1 Report

The authors have provided a an interesting study on AKI, which is a
common clinical condition that appears as a side effect of several
clinical interventions. The authors should highlight this further and
connect this with the possibility that the pathological condition of the
patients undergoing these interventions could deteriorate AKI like
rabdomyolysis does (eg and not restricted to Int J Nephrol. 2019 Nov 26;2019:5010293. doi:
10.1155; Pharmacol Ther. 2017 Dec;180:99-112. doi: 10.1016 etc)

Furthermore, the authors should try and explain in more details why they
selected periostin to study. Although they presented a few evidence,
still the choice seems circumstantial.

Explain the role of water deprivation during the day before the glycerol
injection. Provide reference if this model has been used before.

Try and statistically correlate the level of AKI (as depicted by ig creatinine levels) with
periostin expression and the grades of histopatholocal findings (eg %
tubular damage).

Discuss in more detail the mechanism of AKI including oxidative stress
and how periostin enters the pathways. Refer to already existing prevention strategies and medication using existing literaure (eg and not restricted to 
Toxicol Rep. 2019 May 2;6:395-400. doi: 10.1016; Food Chem Toxicol. 2017 Oct;108(Pt
A):186-193. doi: 10.1016; J Pharm Pharmacol. 2020 Dec;72(12):1822-1829. doi: 10.1111; Proc Natl Acad Sci U S A. 2020 Jul 7;117(27):15874-15883. doi: 10.1073/pnas.2005477117. Epub 2020 Jun 22.; Cell Death Dis. 2021 Feb 26;12(2):217. doi: 10.1038  etc)

Author Response

According to the reviewer’s suggestion, we added Linear regression analysis for % Tubular damage score (H-E staining) in the kidney in Supplement Table 1 and primer sets of mice periostin-1, MCP-1, TNF-a, IL-6, iNOS, ICAM, IL-13, an dIL-10 in Supplement Table 2.

In addition, we added sentences in our paper in response to the reviewer’s suggestion.

Reviewer 2 Report

This is a good, straight forward paper that helps to define the role of Periostin in muscle damage and kidney function. As pointed out by the authors this is important for targeting therapeutics efficiently.

There is a need for more explanation of the experimental model and the results.

1. Why were only male mice used? Does Rhabdomyolysis-Induced Acute Kidney Injury only effect males and not females? Justification for only using male mice is required

2. The methods do not explain how serum was collected and measured for CK and Creatine? More details are required.

In the results you refer to PCR but this is not in the methods. The PCR primers should be included in the methods.

3. WHY did you look at Creatine and not creatinine? Both CK and Creatinine are good markers of injury. Creatine is usually not used as a measure for injury. Please include two references where Creatine is used as a measure of tissue damage.

Also more explanation for the significance of the Creatine vs CK results are required. What does the result mean? What is the physiological basis for this result?

4. In figure 1A you indicate you are sacrificing the mice at day 2 but in the other figures and text you refer to day 3. it would be more accurate to indicate the times in hours, not days.

5. Figure 2: The individual points should be graphed. The error bars are very large for day 3 and I do not see how this could be statistically significant.

There are two minor typos. Line 63. Do not use etc. please list all drugs with references. Line: 96. Space needed between 'technology' and 'fibroblast' and capitalize the 'f'.

Author Response

(The authors gave the same response as above.)

Reviewer 3 Report

The authors explored the role of periostin in acute kidney injury. The study is of some interest but requires modifications.

1. The study was conducted on mice - it has to be reflected in the title. 

2. The authors used glycerol to induce kidney damage. This is described in Methods, however. please provide references for this model. Please also provide proof that indeed glycerol is causing rhabdomyolysis in your study design.

3. The authors provide the information, that mice were sacrificed on day 3 However, in the first paragraph of the Results observation lasting up to day 42 - please explain.

Author Response

(The authors gave the same response as above.)

Round 2

Reviewer 2 Report

All of the reviewers comments were sufficiently addressed.